# Molecular Biocompatibility of a Silver Nanoparticle Complex with Graphene Oxide to Human Skin in a 3D Epidermis In Vitro Model

**DOI:** 10.3390/pharmaceutics14071398

**Published:** 2022-07-01

**Authors:** Marlena Zielińska-Górska, Ewa Sawosz, Malwina Sosnowska, Anna Hotowy, Marta Grodzik, Konrad Górski, Barbara Strojny-Cieślak, Mateusz Wierzbicki, André Chwalibog

**Affiliations:** 1Department of Nanobiotechnology, Institute of Biology, Warsaw University of Life Sciences, 02-786 Warsaw, Poland; ewa_sawosz@sggw.edu.pl (E.S.); malwina_sosnowska@sggw.edu.pl (M.S.); anna_hotowy@sggw.edu.pl (A.H.); marta_grodzik@sggw.edu.pl (M.G.); barbara_strojny@sggw.edu.pl (B.S.-C.); mateusz_wierzbicki@sggw.edu.pl (M.W.); 2Interdisciplinary Division for Energy Analyses, National Centre for Nuclear Research, 05-400 Otwock, Poland; konrad.gorski@ncbj.gov.pl; 3Department of Veterinary and Animal Sciences, University of Copenhagen, 1870 Frederiksberg, Denmark

**Keywords:** silver nanoparticles, graphene oxide, biocompatibility, 3D epidermis, morphology, cytokines

## Abstract

Silver nanoparticles (AgNP) can migrate to tissues and cells of the body, as well as to agglomerate, which reduces the effectiveness of their use for the antimicrobial protection of the skin. Graphene oxide (GO), with a super-thin flake structure, can be a carrier of AgNP that stabilizes their movement without inhibiting their antibacterial properties. Considering that the human skin is often the first contact with antimicrobial agent, the aim of the study was to assess whether the application of the complex of AgNP and GO is biocompatible with the skin model in in vitro studies. The conducted tests were performed in accordance with the criteria set in OECD TG439. AgNP-GO complex did not influence the genotoxicity and metabolism of the tissue. Furthermore, the complex reduced the pro-inflammatory properties of AgNP by reducing expression of IP-10 (interferon gamma-induced protein 10), IL-3 (interleukin 3), and IL-4 (interleukin 4) as well as MIP1β (macrophage inflammatory protein 1β) expressed in the GO group. Moreover, it showed a positive effect on the micro- and ultra-structure of the skin model. In conclusion, the synergistic effect of AgNP and GO as a complex can activate the process of epidermis renewal, which makes it suitable for use as a material for skin contact.

## 1. Introduction

The search for an effective and at the same time biocompatible disinfection system in the time of the COVID-19 pandemic has become very topical [1]. Most biocides are based on alcohols such as ethanol or propanol, used at a concentration of 70–80% in 1 min [2,3]. The basic shortcomings of this solution include, first of all, the disposability of this system, but also the potential toxicity to the skin [4], and even the carcinogenicity of alcohol [5].

The selected nanomaterials could provide an alternative to traditional biocides.

Silver nanoparticles (AgNP) undoubtedly show antibacterial properties [6] against both G(+) bacteria [7] and G(−) [8] as well as viruses [9] and fungi [10,11]. AgNP have a nonspecific and very broad spectrum of antimicrobial activity resulting from the mechanism of their action, especially their unique affinity to the bacterial cell membrane, which causes damage [12].

The basic problems for the wider use of AgNP include difficulties with their standardization and the related lack of sufficient documentation of their biocompatibility. The huge diversification of morphology and the size and purity of AgNP resulting from the production process makes their standardization, registration, and practical application difficult. Their strong tendency to agglomerate and the protein crown effect also appear to be key difficulties which reduce their antimicrobial effectiveness [13] and makes it difficult to predict. Another and fundamental reason for doubts about the therapeutic use of AgNP is their toxicity. The use of silver nanoparticles for disinfecting a room, clothes, and everyday objects is associated with their contact with the skin. The use of AgNP as an antimicrobial agent, and in particular as a personal antiseptic agent and also as an additional safety factor applied to face masks [14], is becoming more relevant in the time of the COVID-19 pandemic. Therefore, research into AgNP interactions with human skin seems to be indispensable and urgent.

The behavior of AgNP in contact with skin and in the body has been studied both in vitro and, to a lesser extent, in vivo. Larese et al. [15] documented a low absorption of AgNP coated with polyvinylpyrrolidone through the skin and its presence in the stratum corneum in an in vitro study. Moreover, skin damage favored AgNP penetration. In in vivo studies on pig skin, penetration of AgNP into the stratum corneum of the epidermis was also observed after a 14-day contact with a colloid of AgNP [16]. After penetrating the skin, the nanoparticles can be distributed to various organs, but ultimately, they are most often removed from the body. In AgNP clearance kinetic studies, removal of AgNP from non-barrier-free organs (brain, testes) was observed after up to four months [17,18]. However, these studies were conducted after oral administration, blood administration, and AgNP inhalation. Similar mechanisms probably apply to nanoparticles absorbed by the skin, although the skin is the most effective protection system of the body. Nevertheless, the optimal solution to this problem would be a modified AgNP preparation that would reduce the mobility of AgNP into the skin.

Tseng et al. [11] argue that the antimicrobial efficacy of the AgNP colloid is at a level that is not harmful to humans. Nevertheless, the search for a method that would improve the performance of AgNP, reduce its absorption through the skin, and completely eliminate the potential cytotoxicity and genotoxicity of nanoparticles is timely, considering the search for new decontamination options in the time of the COVID-19 pandemic. It seems that the attachment of AgNP to the graphene oxide (GO) flake platform would reduce the absorption of nanoparticles by the skin, reduce their toxicity, and extend the action of AgNP over time through slow ion release and very slow migration of AgNP from GO flakes.

Why is GO an ideal carrier for AgNP? GO is a structure up to several nanometers thick, depending on the number of flake layers. It is transparent, flexible, durable, and with the presence of numerous -OH, -COOH, =O groups, it is susceptible to self-assembly and functionalization by other compounds. First of all, GO seems to be a biocompatible material. Studies by Pelin et al. [19] on keratinocytes have shown that high concentrations and long exposure to GO is cytotoxic to cells, while low concentrations of GO are biocompatible. Moreover, studies in a human skin model (SkinEthic™—Reconstructed Human Epidermis) also showed no toxicity of GO, as assessed by the MTT test, its histological picture, and the secretion of proinflammatory cytokines [20]. However, contamination with chemical residues from the production process increased the toxicity of GO. There are relatively few tests of GO’s toxicity to skin, compared to the toxicity of GO to the lungs, eyes, or when it is administered into the blood and orally [21]. The dermal route of exposure is extremely important, which additionally inspires the studies we have undertaken.

Due to its low toxicity and remarkable physical properties, GO seems to be an excellent carrier for the AgNP biocidal substance. Earlier studies by our team investigated the toxicity of a foil dressing with GO applied and encrusted with AgNP [22,23]. Interestingly, the introduction of GO into this complex reduced the toxicity of silver without reducing the antibacterial properties of the composite. These studies were performed on in vitro models of fibroblasts and HUVECs and in ovo—chicken embryo chorioallantoic membrane. Other model studies on the toxicity of the AgNP-GO complex carried out on zebrafish embryos and caprine fetal fibroblasts documented the dose-dependent nature of the toxicity of this material [24,25]. Interesting studies on the toxicity of the AgNP-GO in vitro were carried out on macrophages. The results indicated a more toxic synergistic effect of AgNP-GO compared to the single action of GO and AgNP [26]. Although the AgNP-GO nanocomposite was internalized by macrophage cells to a lesser extent, it stimulated oxidative stress more intensely. However, although the external application of the AgNP-GO complex is primarily related to the contact and interaction of the complex with the epidermis, the contact of this material with macrophage cells present in the dermis cannot be ruled out.

The aim of the research undertaken here was to assess the biocompatibility and toxicity of the composite of AgNP with GO (AgNP-GO complex) in studies on the 3D human skin model. There was particular emphasis on the assessment of the micro- and ultra-structure, potential pro-inflammatory activity, and genotoxicity.

## 2. Materials and Methods

### 2.1. Characterization of Experimental Factors

The hydrocolloid of AgNP was obtained from the Nano-koloid company (Warsaw, Poland). it was produced with a proprietary (Polish patent no. 3883399), nonexplosive, high voltage method using high purity metal (99.9999%) and high-purity demineralized water. The concentration of AgNP used in the experiment was 25 mg/L.

The GO flakes dispersed in the aqueous solution at the 4 mg/mL concentration were purchased from the NanoPoz company (Poznan, Poland). They were produced by a modified Hummers method and have a 36% concentration of oxygen. The GO solution was suspended in ultra-pure water to give the final concentration of 5 mg/L that was used in the experiments and for other measurements.

The third factor was the combination of AgNP and GO. This was obtained as a result of a self-organization process with the use of a sonication method according to Wierzbicki et al. [23] (ultrasonic bath, Bandelin Electronic, Berlin, Germany).

All experimental factors (AgNP, GO, and AgNP-GO complex) were characterized and analyzed using the following methods:

ζ-Potential measurements and size distribution: The ζ-Potential of AgNP, GO, and GO-AgNP were measured by the laser dynamic scattering electrophoretic method using the Smoluchowski approximation with a ZetaSizer Nano ZS model ZEN3500 (Malvern Instruments, Malvern, UK). All measurements were performed in triplicate after stabilization at 25 °C and 30 min after the sonication process. The hydrodynamic diameter of the nanoparticles in water and their size distribution were measured with dynamic light scattering (DLS) using a Nano-ZS90 Zetasizer (Malvern Instruments, Malvern, UK).

Transmission Electron Microscopy: The shape and size of the AgNP, GO flakes, and the complex were examined using transmission electron microscopy (TEM) TEM JEM-1220 (JEOL, Tokyo, Japan) at 80 kV on equipment with a TEM CCD Morada 11 megapixel camera (Olympus Inc., Tokyo, Japan). The samples were prepared by placing hydrocolloids onto Formvar-coated copper grids (Agar Scientific Ltd., Stansted, UK) and air drying before TEM imaging.

pH measurement: The measurement of pH stability of the ultra-pure water (solvent) and investigated nanomaterials was performed in triplicate, at 21 °C, with the CP-411 pH meter (Elmetron, Zabrze, Poland). The analysis was repeated at the selected points of time (0, 1, 3, 24, 48 h).

Fourier-transform infrared spectroscopy measurement (FTIR): The FTIR spectra of AgNP-GO complex were performed according to the method described by Sosnowska et al., (2021) [27], using a Perkin Elmer System 2000 instrument (PerkinElmer, Inc., Waltham, MA, USA) operated by Pegrams 2000 software (PerkinElmer, Inc., Waltham, MA, USA) in the range of 400–4000 cm^−1^. Dried samples of AgNP-GO were milled with potassium bromide (KBr) crystals in the ratio of 1:300 mg in a laboratory mill (Specac Ltd., Orpington, UK). Pellets were prepared using 8 tons of pressure for 2 min. A total of 25 scans were carried out for every sample.

### 2.2. Model Description and Experimental Design

The EpiDerm™ Skin irritation model (EPI-200-SIT) was obtained from MatTek In Vitro Life Science Laboratories (Bratislava, Slovakia; ISO 9001:2008 certified), part of MatTek Corporation (Ashland, OR, USA). This model is patented and proven by the ECVAM Scientific Advisory Committee as an in vitro 3D model system for chemical, pharmaceutical, and skin care product testing [28]. EpiDerm™ consists of normal, human-derived, epidermal keratinocytes cultured on specially prepared tissue culture inserts, and it exhibits human epidermal tissue structure and cellular morphology.

The main goal of the experiments performed was to validate the biocompatibility of experimental factors. Therefore, the model samples (*n* was a number that varied depending on the analysis performed, but it was not less than 3 per group) were divided into 5 groups: negative control (NC), treated with Dulbecco’s phosphate-buffered saline (DPBS); positive control (PC), treated with 5% sodium dodecyl sulfate (SDS); and 3 experimental groups with AgNP (concentration 25 mg/L), with GO (concentration 5 mg/L), and the complex with the same concentrations of AgNP and GO. The concentrations used in the experimental groups resulted from previous experience (own data, unpublished). All of the necessary media (such as SDS, DPBS, assay medium) were supplied by the manufacturers (MatTek Corporation, Ashland, OR, USA).

The experiment was conducted in accordance with the criteria set in OECD TG439, as recommended by the manufacturer of the validated SIT test and data from specific literature [28]. Briefly, at day 0 (when we received the model), the samples were pre-prepared as indicated in the MatTek protocol. Next, after 18 h, the assay medium was renewed, and the 30 µL/insert of tested factor or DPBS or SDS were added on the top of the inserts containing the tissues. Then, a nylon mesh was placed on the surface, and the model was incubated for 35 min at 37 ± 1 °C, 5 ± 1% CO_2_, 90% ± 10% relative humidity and then for 25 min in the sterile hood. After a total of 1 h of tested substance exposure, the tissues were washed with the sterile DPBS, dried, and transferred to a new plate with fresh assay medium. The model was next incubated (37 ± 1 °C, 5 ± 1% CO_2_, 90% ± 10% relative humidity) for 24 h. At day 2, the medium was collected to perform IL-1α analysis. Next, the tissues were removed to fresh assay medium and incubated at the conditions described previously for 18 h. At day 3, inserts were correctly fixed, with the exception of samples intended for the viability test. Further proceedings depended on the planned analyses.

### 2.3. MTT Viability Test

At the 3rd day of an experiment, the MTT test was performed to determine the viability of the tissues after nanocomponent treatments. The EpiDerm™ inserts (*n* = 3/group) were removed to a new plate with MTT medium in each well (MatTek Corporation, Ashland, OR, USA) and incubated in an incubator for 3 h. Next, the inserts were washed with DPBS and immersed in isopropanol (also delivered by MatTek). After 3 h at room temperature with gentle shaking on a plate shaker, aliquots of the blue formazan solution were transferred into a 96-well flat bottom microtiter plate in triplicate. Measurement of color intensity, after dehydrogenase conversion of MTT present in cell mitochondria into a blue formazan salt by viable cells with an active metabolism, was performed at an absorbance of 570 nm (Infnite^®^ 200 PRO microplate reader with i-control™ software (Tecan Group Ltd., Männedorf, Germany)). The results (relative tissue viability) were documented and calculated according to the following formulas:Relative viability TS (%) = [ODTS/Mean of ODNC] × 100
Relative viability NC (%) = [ODNC/mean of ODNC] × 100
Relative viability PC (%) = [ODPC/mean of ODNC] × 100
where the abbreviations have the following meanings: ODTS—optical density of tested substance; ODNC—optical density of negative control, and ODPC—optical density of positive control.

### 2.4. Determination of IL-1α Concentration in Medium

The medium for interleukin-1α (IL-1α) analysis was collected 24 h after the treatment (at day 2nd) and frozen at −80 °C. The concentration of this proinflammatory cytokine was detected with a colorimetric Human IL-1α ELISA Kit (Abcam, Cambridge, MA, USA) by strictly following the assay kit’s instructions. The measurement of the absorbance of the HRP/TMB colored base signal at 450 nm, from standards control and samples, was detected using an Infnite^®^ 200 PRO microplate reader with I-control™ software (Tecan Group Ltd., Männedorf, Germany). Calculations were performed as described in the manufacturer’s protocol.

### 2.5. Activity State of the Inflammatory Cytokines

To evaluate the impact of experimental factors on the protein expression of proinflammatory cytokines in EpiDerm™, a Human Inflammation Antibody Array, a membrane for 40 targets (Abcam ab134003, Cambridge, MA, USA) test was performed. These membranes gave results in duplicate. Tissues (*n* = 3/group) for this analysis were frozen (−80 °C) at the 2nd day of an experiment. The analysis was performed directly according to protocol. Briefly, protein from samples was extracted and standardized with the BCA method according to the producer’s instructions. Array membranes were blocked with blocking buffer from a kit for 30 min at room temperature (RT). After aspiration, 250 µg of total protein from each sample, diluted in 1 mL of blocking buffer, was added per array membrane. The samples were incubated overnight at 4 °C. Next, after the washing step, antibodies conjugated with biotins were added and incubated for the next 24 h at 4 °C. Then, after a washing step, the membranes were incubated with streptavidin conjugated with horseradish peroxidase for 2 h at RT. Finally, the membranes were visualized after the addition of the provided horseradish peroxidase substrate using a ChemiDoc imaging system (Bio-Rad, Hercules, CA, USA). Densitometric analysis of the signal was performed using the Protein Array Analyzer tool for ImageJ software (Research Services Branch, National Institute of Mental Health, Bethesda, MD, USA) [29]. To normalize array data, the calculations were performed in line with the manufacturer’s instructions.

### 2.6. DNA Damage Evaluation by Detecting 8-Hydroxy-2-Deoxyguanosine Concentration (ELISA Kit)

The level of oxidative stress indicator in EpiDerm™ was measured 24 h after treatment with 8-hydroxy-2’-deoxyguanosine (8-OHdG) by an enzyme-linked immunosorbent assay (ELISA) kit (no. ab201734; Abcam, Cambridge, UK), following the instructions of the manufacturer. Absorbance values were measured on a microplate reader at 450 nm (Infnite^®^ 200 PRO microplate reader with i-control™ software (Tecan Group Ltd., Männedorf, Germany)). The 8-OHdG concentration was calculated from the standard curve.

### 2.7. Visualization of the EpiDerm™

The four types of the EpiDerm™ visualization were performed.

Two methods of surface visualization were used: At first, digital photos of the EpiDerm™ surface 24 h after treatment without fixation were obtained using a stereo microscope (SZX10, CellD software v3.1; Olympus Corporation, Tokyo, Japan). Also, a scanning electron microscope (SEM; Quanta 200, FEI, Hillsboro, OR, USA) was applied. Tissues imaged in SEM were first fixed in 2.5% l-glutaraldehyde in PBS and then processed as described previously [30].

Histology of EpiDerm™ tissues: 24 h after treatment, tissues exposed to all experimental factors and from negative/positive control groups were fixed in 10% paraformaldehyde overnight at RT. After the standard MatTek histology characterization sample preparation procedure (dehydrated, paraffin embedded, sectioned, and HE stained), the slides were observed and recorded using a Nikon Eclipse Ni light microscope with a Nikon DS-Fi3 camera (Nikon Corporation, Tokyo, Japan) using 10×, 20×, and 40× magnification. Nikon Elements software was used to process images. We established three parameters to prove the proper activity and functionality of the model after treatment: the number of nuclei in the stratum corneum, the thickness of the corneal layer, and total thickness of EpiDerm™. All measurements were performed in a randomly selected visual field (*n* = 12 per group).

Ultrastructure visualization in cross-sections of EpiDerm™ (TEM Analysis): Similar to the previous visualizations, 24 h after experimental treatment tissues were removed from the cell culture inserts and immediately cut into four equal parts, then fixed in a 2.5% glutaraldehyde solution (Sigma-Aldrich, St. Louis, MO, USA) in 0.1 M PBS (pH 7) overnight. Then, the samples were washed in the PBS and transferred to a 1% osmium tetroxide solution (Sigma-Aldrich, St. Louis, MO, USA) in 0.1 M PBS (pH 7) for 1 h. Next, they were washed in distilled water, dehydrated in ethanol gradients, and impregnated with epoxy embedding resin (Fluka Epoxy Embedding Medium Kit; Sigma-Aldrich, St. Louis, MO, USA). After 24 h, the samples were embedded in the same resin and baked for 24 h at 36 °C; then, they were transferred to a 60 °C incubator and baked for a further 24 h. Ultrathin sections (50 nm) obtained with an ultramicrotome (Ultratome III; LKB Products, Vienna, Austria) were transferred onto TEM grids (Formvar on 3 mm 200 Mesh Cu Grids, Agar Scientific, Stansted, UK). Sections were contrasted using uranyl acetate dihydrate (Sigma-Aldrich, St. Louis, MO, USA) and lead (II) citrate tribasic trihydrate (Sigma-Aldrich, St. Louis, MO, USA) and examined by TEM JEM-1220 (JEOL, Tokyo, Japan).

### 2.8. Statistical Analysis

Data were analyzed by one-way ANOVA. The differences were evaluated by Tukey’s HSD test. Statistical analysis was performed using IBM SPSS statistics, v25 (SPSS Inc., Chicago, IL, USA). Data are expressed as the mean and standard error of mean. For all tests, statistical differences with *p* ≤ 0.05 were considered significant.

## 3. Results

### 3.1. The Characterization of the Experimental Factors

To evaluate the morphology of the nanomaterials and their interactions in the AgNP-GO complex, the TEM analysis was performed (Figure 1A). Visualization of AgNP showed an irregular shape without sharp edges, and a single particles’ size range was between 40–80 nm. The GO flakes were uniform, forming 1–3 layers, and they were angular with rather sharp edges. The diameter of single flakes was from 1.5 to 3 μm. The most interesting occurrences were images representing the AgNP-GO complex, which appeared to have adhesion of silver nanoparticles to the surface of the GO flakes (yellow arrows at the graph). Additionally, measurement of the ζ-potential showed that all experimental factors in aqueous solution were characterized by high stability, and there was no tendency to agglomerate (Figure 1B). Their ζ-Potential values were as follows: AgNP, −27.87 mV (±0.71 mV); GO, −49.25 mV (±2.97 mV); and AgNP-GO complex, −32.10 mV (±0.82 mV). Moreover, dynamic light scattering (DLS) analysis was performed. The Z-average sizes of the experimental factors were established: 78.2 nm (±22.29 nm) for AgNP, 965 nm (±34.12 nm) for GO, and 893 nm (±7.57 nm) for complex. Their size distributions are presented in Figure 1C. The pH measurement, performed in the selected points of time, resulted in, at point 0: 6.34 (±0.12) (water), 6.45 (±0.08) (AgNP), 6.42 (±0.1) (GO), 6.52 (±0.05) (AgNP-GO). With time, all measurements did not change significantly and were in the pH range of 6.5–6.8 at the last point of measurement.

Additionally, the FTIR spectra of the AgNP-GO complex are presented in Figure 2. We can observe the presence of a 3442 cm^−1^ broad band associated with -OH groups. The 2810 cm^−1^ and 2768 cm^−1^ bands indicate CH groups. The 1720 cm^−1^ band in the spectrum is related to the C=O group. A band of 1619 cm^−1^ indicates the groups C=C, derived from GO. Bands found at 1359 cm^−1^ and 1054 cm^−1^ are associated with the CH groups.

### 3.2. Assessment of Cytotoxicity of the Experimental Factors

To compare the effect of AgNP, GO, and the AgNP-GO complex on EpiDerm™ viability, which is the basic parameter determining potential cytotoxicity of the experimental factors, MTT reference tests were performed and the outcomes are presented in Figure 3A. The results obtained showed no differences after the GO and complex treatment, when compared to the negative control (NC) (DPBS treatment), but we observed a significant decrease in the relative absorbance value after AgNP treatment (−25.2%). However, this result is still above 50% of the mean viability of the negative controls (the level is marked with the blue line at the graph).

### 3.3. Genotoxicity Potential of Nanomaterials

The 8-OHdG level was measured as an oxidative DNA damage marker. The results were evaluated for all groups except the positive control (PC) (SDS treatment) (Figure 3B). The 8-OHdG levels were not significantly different among the AgNP-, GO-, AgNP-GO-treated, and NC (DPBS)-treated groups.

### 3.4. Effect of Nanomaterials on Proinflammatory State

The effect of the tested factors on the inflammation state of EpiDerm™ inserts was evaluated both by analyzing the concentration of IL-1α released into the culture medium 24 h after contact with the tested agents and by testing the protein expression state in the tissues on the 3rd day of the experiment.

As we can observe in Figure 3C, there was no significant effect (neither increase nor decrease) of the experimental factors on the concentration of IL-1α released into culture medium in which the inserts were immersed. The results were similar to those observed in NC (DPBS treatment).

To complete the data on the potential inflammation state, an analysis of 40 cytokines’ expression was performed using a protein array in the investigated model. In this analysis, PC (SDS treatment) was not considered. The locations of particular proteins on the membrane are presented in the table (Figure 3D). The presented results (Figure 3C) show the changes (increase or decrease) in the expression of the tested proteins in relation to the NC results. AgNP treatment induced an increase in protein level expression of the following cytokines: interferon gamma-induced protein 10 (IP-10) (also known as C-X-C motif chemokine ligand 10 (CXCL10)), tumor necrosis factor β (TNFβ), interleukin 3 (IL-3), and interleukin 4 (IL-4). It is worth noting that the latter two cytokines were present only in this group. At the membranes representing the GO group, we observed increased activity of IL-1α, IP-10, TNFβ, Tissue inhibitor of metalloproteinases 2 (TIMP2), and macrophage inflammatory protein 1β (MIP1β), also known as chemokine (C-C motif) ligands 4 (CCL4). While in the samples after complex treatment, a 3-fold decrease in MIP1β expression was noticeable when compared to the NC, and there was no expression of IP-10. Moreover, an increase in TNFβ, TIMP2, and IL-1α were observed. The cytokine platelet-derived growth factor BB (PDGF-BB), which was clearly expressed in the control group, was not detected in any of the studied groups. In turn, on activation, cytokine regulated normal T cells expressed, and secreted RANTES, also known as chemokine (C-C motif) ligand 5 (CCL5), was found in all groups except in the samples from NC.

### 3.5. Visualization of EpiDerm™

The EpiDerm™ model is cultured at the air–liquid interface to form a multi-layer. It is a 3D structure consisting of organized and proliferative cells which are mitotically and metabolically active. In order to comprehensively evaluate the examined tissues, we used four types of visualization.

To assess the outer surface that has been in direct contact with the tested substances, we used a stereoscopic microscope (Figure 4A) and SEM technology (Figure 4B). As we can observe in the macro scale, there is a disruption in the continuity of the EpiDerm™ surface only after SDS treatment (PC), whereas the use of tested agents did not affect the uniformity of the corneal layer of the samples, and the results after AgNP, GO, and complex treatment are similar to NC (DPBS). Additionally, SEM analysis confirms no toxic effect of the tested substances, and they even seem to support the condition of the top layer of the EpiDerm™. After AgNP and especially after AgNP-GO complex treatment, the outer layer of the EpiDerm™ was very smooth without any symptoms of exfoliation. Meanwhile, in samples treated with GO, we observe a very slight keratinization process of the top layer (white arrow), similar to the group treated with DPBS where we see a fairly homogeneous structure with single signs of keratinization and the initial stage of exfoliation (white arrows). Pictures from the PC group show a typical image for irritating substances, where we can see the lack of cohesion of the top layer; yellow arrows indicate where tissue is broken. All of the outer layer of the tissue is macerated by the action of a strong detergent (5% SDS).

To investigate the condition of all layers of the EpiDerm™, we performed visualizations of samples in a cross section with standard HE staining (Figure 5). The EpiDerm™ is complex structure. It is composed of stratum corneum, stratum granulosum, and spinous and basal layers (Figure 5A). We observed no significant irritant effect of the experimental factors on the EpiDerm™ structure (Figure 5B). However, after AgNP application, a few places with local thickening of the stratum corneum (Figure 5B, yellow arrow), which is a common response to dermal application of chemicals, were noticeable. Similar to samples from the NC group, all layers were properly stained, which makes them easy to distinguish. In addition, there are no inflammatory lesions in any of the layers. When we look at the representative images of samples from the PC (SDS) group, which show a high level of damaged stratum corneum (black arrows) and other defects in lower layers that testify to necrosis process (yellow arrows), then we can clearly state that AgNP, GO, and the complex show no irritating effects on the tested structures.

Stratum corneum is a layer constructed from keratin anucleate cells. However, we can observe some nuclei in this stratum (they are marked with red arrows in Figure 5B). Comparing the average number of nuclei in the corneal layers between groups, we can state that number of nuclei in the group treated with the AgNP-GO complex is significantly lower when compared to the group with DPBS treatment (5.8 compared to 14.4, respectively) (Figure 5C). There were no statistical differences between AgNP and GO groups; however, there was a noticeable downward trend.

We also evaluated the thickness of the corneal layer (Figure 5D) and the total thickness of EpiDerm™ (Figure 5E). The measurement limits are marked on the diagram with yellow and green double headed arrows, respectively (Figure 5A). The most compact and homogeneous morphology of a corneal layer and, at the same time, the thinnest was a characteristic of the samples after the application of the AgNP-GO complex. This diameter decrease was statistically significant compared to the NC (DPBS) (37 µm vs. to 45.3 µm, respectively). In turn, the average total thickness of the EpiDerm™ was significantly lower both after AgNP and AgNP-GO complex treatment, when compared to NC, and the values were 100 µm (AgNP), 98 µm (complex), and 111 µm (DPBS), respectively. No differences between groups were observed after GO treatment.

A detailed observation of EpiDerm™ with TEM showed no pathological changes between groups in the ultrastructure (Figure 6). The investigation was mainly concerned with the cell layers and their specific structures. We did not observe any nanoparticles inside the cells after AgNP and AgNP-GO treatment. The most compact cell layers, with visible strong keratin fibers (blue arrows) and the presence of multiple lipid droplets (yellow arrows), were observed in the samples from the AgNP-GO group.

## 4. Discussion

The physical characteristic of the tested AgNP-GO complex, including visualization, size and shape analysis, zeta potential value, and the FTIR spectrum, was focused on to determine the stability of the tested complex. The presented spectrum from FTIR analysis is a characteristic spectrum for GO [27,31] and shows the presence of -OH groups bound to the AgNP surface. The spectrum does not indicate the formation of covalent bonds between GO and AgNP; moreover, the hydrogen bonds are probably also not dominant due to the lack of a broad peak around 3000 cm^−1^. Similar observations regarding the spectrum of the GO complex and Ag nanoparticles were observed by Lange et al. (2022) [32]. Furthermore, there were no significant differences between the FTIR spectrum and the spectra of GO and Ag-GO complex. However, the conducted research might indicate a possible interaction of Ag with hydrogen-bonded O-H groups.

The EpiDerm™ model is grown at an air–liquid interface and forms a multilayer structure consisting of the epidermal cells of the stratum corneum, the granular layer, the spinous layer, and the basal layer. Although this structure reflects the correct structure of the natural human epidermis, the model is devoid of cells associated with keratinocytes, such as melanocytes, Langerhans cells, or Merkel cells. Nevertheless, this model is considered optimal for in vitro testing of potential skin toxicity [33,34,35]. In order to confirm the reliability of the method, a PC (SDS treatment) was introduced in accordance with the test protocol [33] and it gave negative results in all parameters tested, clearly confirming the correctness of the method used. In order to comprehensively assess toxicity, we used various methods for its assessment, namely, the metabolic response of cells (MTT test), immune response (40 cytokines), genotoxicity (concentration of 8-hydroxy-2’-deoxyguanosine), and four types of visualization. 

In order to compare the potential irritating effects of the AgNP-GO complex as well as its components (AgNP and GO) on the 3D EpiDerm™ skin model, the MTT reference test was performed. It is considered suitable for this type of research [36]. Exposure of the EpiDerm™ to AgNP slightly decreased the activity of cellular metabolism, indicating a slight irritant effect. However, according to the EU classification (R38/Category 2 or no class), an agent is considered to be an irritant if the mean relative viability of the three individual tissues exposed to the test chemical is reduced below 50% of the mean viability of the negative control [28]. Therefore, none of the factors examined can be considered to be irritating. However, even slight toxicity can activate inflammatory processes and even immune stress [37]. Therefore, the next step in our research was to evaluate the secretion of IL-1α by EpiDerm™ cells into the medium.

IL-1α, a member of the IL-1 family of cytokines, is also secreted by “non-immune” cells, including keratinocytes [38]. In addition, keratinocytes are critical in regulating skin inflammatory responses, and extracellular IL-1α release is considered a key marker of chemical skin irritation [39,40]. This observation was also confirmed in our study due to the very high level of IL-1α after SDS treatment. The use of the AgNP-GO complex as well as the individual components of AgNP and GO did not increase the concentration of IL-1α in the medium, which would indicate their biocompatibility. Other studies also confirm the lack of pro-inflammatory effects from moderate doses of AgNP on the skin or epidermis [16,41], and even a reduction in inflammation (edema response) has been observed with AgNP [42]. IL-1α is secreted only by necrotic cells and not by apoptotic cells [40]. Therefore, although a decrease in viability (MTT test) was associated with apoptosis, IL-1α may not be secreted. Regarding GO, other studies also confirm that GO does not have a pro-inflammatory effect on the skin (it does not increase IL-1α levels) [20,43]. However, we wanted to confirm the obtained results by examining the protein expression of 40 other chemokines. The test showed that only 11 out of 40 different chemokines were altered by the factors tested. Overall, of these 11 cytokines, most are pro-inflammatory or involved in the inflammatory process (IL-1α, IP-10, MIP1β, IL-3, IL-4, PDGF-BB, and RANTES) and two are anti-inflammatory (TIMP2, TNF-β).

The key pro-inflammatory cytokine (IL-1α) secreted by keratinocytes was increased due to contact with GO. Other authors also observed an increase in IL-1α secretion by HaCaT cells upon contact with GO [19]. However, in the studies on the 3D skin model, we did not observe the irritating effect of GO, including expression of IL-1α [20]. The AgNP-GO complex did not decrease this pro-inflammatory effect of GO. Thus, the complexation of AgNP and GO did not reduce the biological activity of GO in the proinflammatory direction. It should be emphasized, however, that GO increased the synthesis of IL-1α in 3D skin cells. However, this was not reflected in the level of IL-1α secreted into the culture medium. Thus, this mechanism induced processes related to the fate of the cell rather than the promotion of inflammation within its environment. It can be assumed that IL-1α, and especially its propiece, may have been involved in the apoptosis [44] of damaged cells. Furthermore, contrary to the presence of IL-1α in the culture medium, in the AgNP group, no expression of this protein was found in the EpiDerm™ tissue. Moreover, the analysis showed the presence of IL-1α in the tissue treated with GO and the AgNP-GO complex. It can be hypothesized that the AgNP, with much smaller structures than GO and the AgNP-GO complex, could interact much faster with proteins exposed outside the cell and with intracellular proteins. This could delay the synthesis and secretion of extracellular IL-1α; so, this process was in the initial phase (synthesis) in the GO and AgNP-GO groups, and it was in the phase of extracellular cytokine secretion in the AgNP group.

All of the nanostructures used increased the expression of the TNF-β protein. TNF-β belongs to the family of TNF proteins produced by activated macrophages but also expressed in fibroblasts and keratinocytes [45]. However, chondrocyte studies have shown that TNF-β is involved in microenvironmental inflammation and activates the inflammatory response [46] but also TNF-β stimulates the growth of human dermal fibroblasts participating in the process of wound healing [47] and it is believed to be an anti-inflammatory cytokine involved in inhibiting the production of pro-inflammatory mediators [48]. Thus, not only AgNP and GO but also AgNP-GO could slightly mobilize factors responsible for inhibition of inflammation. Moreover, the production of TNF-β can be activated by primary proinflammatory cytokine IL-1α [46]. This may suggest the restorative nature of the contact between AgNP, GO, and AgNP-GO and the epidermis.

Like TNF-β, the expression of the RANTES also increased in the AgNP, GO, and AgNP-GO groups. The secretion of RANTES, a chemotactic cytokine, by keratinocytes is regulated by inflammatory cytokines such as IL-1β, TNF-α, IL-4, and IL-13 [49]. In our experiment, we observed expression of IL-1β in GO and AgNP-GO groups as well as expression of IL-4 in the AgNP group, coinciding with the presence of RANTES, but this does not support the thesis about the anti-inflammatory nature of AgNP-GO.

However, expression of MIP1β (CCL4) chemokine can confirm the non-toxicity and even anti-inflammatory properties of the AgNP-GO complex. The pro-inflammatory chemokine MIP1β, elevated, inter alia, myocardial infarction [50], was increased by GO, while it was significantly reduced by the AgNP-GO complex, as well as when compared to the control group. However, the role of GO is not unequivocally pro-inflammatory. This is due to the expression of another chemokine, TIMP-2. TIMP-2 is a member of the TIMP protein family, and it has multiple effects on dermal tissue. For example, it accelerates wound healing by enhancing the proliferation and migration of epidermal keratinocytes and dermal fibroblasts. In addition, it is a physiologic inhibitor of matrix metalloproteinases [51] and collagen synthesis [52]. The result of 2-fold increased expression of TIMP-2 after GO application would suggest that GO, regardless of the presence of AgNP, slightly activates skin repair mechanisms.

An example of a diametrically different action of the complex compared to its components is the expression of the chemokine IP-10. IP-10 (CXCL-10), a key pro-inflammatory cytokine involved in the recruitment of leukocytes to inflammatory sites [53] was elevated under the influence of AgNP and GO treatment. Interestingly, the level of this cytokine was suppressed by the AgNP-GO complex. Moreover, IP-10 concentration was higher in the NC (DPBS treatment) group than in the AgNP-GO group. This suggests the involvement of some bonds in the AgNP and GO complexation process and the lack of their accessibility to the skin in the AgNP-GO complex. This documents the very high neutrality of the AgNP-GO complex towards the skin.

PDGF-BB (platelet-derived growth factor BB) protein is produced by various types of skin cells, including keratinocytes. It is synthesized locally in response to damage [54,55,56]. However, even low PDGF-BB expression is also observed in cells not exposed to the toxic agent [55]. Interestingly, even this low level of PDGF-BB expression was not observed under the influence of AgNP, GO, or the AgNP-GO complex in comparison to the control. Undoubtedly, this indicates the lack of toxicity of the experimental factors when used at the level of 25 mg/L (AgNP) and 5 mg/L (GO).

AgNP influenced the activation of the pro-inflammatory IL-3 and IL-4 synthesis. However, a different function of these cytokines can be considered. The interleukins IL-3 and IL-4 are also involved in hematopoiesis. AgNP are used to coat dressings as a wound healing agent. They act not only as an antibacterial agent but also to reduce local inflammation via cytokine modulation, and furthermore they reduce scarring [57]. This effect may be related to the local stimulation of angiogenesis by AgNP. This was observed in our studies on chicken embryo as an increase in the expression of VEGF-A and FGF-2 and an improvement in the structure of blood vessels [58,59]. However, this hypothesis requires confirmation in angiogenesis studies.

Summarizing the analysis of the expression of 40 cytokines under the influence of the studied factors, it can be undoubtedly stated that AgNP and GO are not indifferent to the epidermal tissue. They induce the production of proteins of both a pro-inflammatory and an anti-inflammatory nature. However, what is most interesting is that combining AgNP and GO into the AgNP-GO complex reduced the pro-inflammatory properties of AgNP and/or GO by reducing the expression of IP-10 (expressed in AgNP and also in GO groups) as well as MIP1β (expressed in GO group) and IL-3 and IL-4 (expressed in AgNP group).

Interesting studies on the toxicity of the GO and AgNP complex in vitro were carried out on macrophages. They indicated a more toxic synergistic effect of GO-AgNP compared to the single action of GO and AgNP [26]. Although the GO-Ag nanocomposite was internalized by macrophage cells to a lesser extent, it stimulated oxidative stress more intensely. It should be emphasized that AgNP, GO, and AgNP-GO were not found to be genotoxic in our studies, which may indicate a low level of oxidative stress and ROS generation at the physiological level. Thus, the toxicity of the AgNP-GO complex is critically dependent on the biological model and practically on the method of administration. In addition, Chen et al. [60] documented that in cell line studies, the toxicity of AgNP increases compared to studies on the 3D skin model. This is due to the hindered penetration of cells into the 3D skin structure. In addition, previous studies by our team investigated the toxicity of a foil dressing to which the AgNP-GO complex was applied [22,23]. The introduction of GO to the complex reduced the cytotoxicity of silver without reducing its antibacterial properties. Moreover, the results of studies on zebrafish suggest that GO is a preferred transport platform due to its low toxicity, and it also reduces AgNP toxicity [24].

The morphological evaluation of the epidermal model was carried out at the microstructure and ultrastructure levels using several methods (namely, inverted and histological microscopy and scanning and transmission electron microscopy) which allowed for a comprehensive analysis of the state of the epidermis. First of all, none of the visualization methods, especially TEM observation, showed the presence of AgNP or GO or their aggregates. This is contrary to other studies on AgNP [15,61,62] and nanoparticles of GO [15,56]. However, the main difference was the exposure time, which in our studies on a human skin model was 1 h.

By analyzing the surface condition of the stratum corneum, it can be emphasized that AgNP and the AgNP-GO complex increased smoothness and decreased surface roughness. Stratum corneum, the top layer of the epidermis, is made up of keratin-containing non-nucleated cells arranged in a basket-like form. The study of the physiological response of the skin is confirmed by the assessment of its surface structure (SEM). The observation of EpiDerm™ ultrastructure clearly showed very slight changes in the skin structure in the form of its keratinization under the influence of GO. On the other hand, the action of AgNP, as well as AgNP-GO, improved the image of the epidermis surface. This was observed as an increase in its smoothness and lack of exfoliation. Moreover, the smoothest EpiDerm™ surface was characterized the action of the AgNP-GO complex. The stratum corneum is composed of dead cells filled with keratin, which separate from each other, undergoing exfoliation which allows for the constant renewal of the skin. It can be assumed that AgNP influenced faster exfoliation of dead cells, and this is to the advantage of younger cells that are capable of more intensive keratin synthesis. This hypothesis is supported by the MTT results which indicated a slight irritating effect of AgNP. This effect may suggest an action similar to peeling. On the other hand, the skin surface structure in the AgNP-GO group was also smoother compared to the control group, and it did not affect the MTT test. When analyzing the number of nuclei in stratum corneum cells, a significant reduction of their number could be observed in the AgNP-GO group. This is associated with a reduction in the thickness of the stratum corneum, also characteristic of the AgNP-GO group.

When analyzing the histological image (HE staining) of the stratum corneum, a slight decrease in its thickness can also be observed when exposed to AgNP-GO compared to the NC group and the AgNP and GO groups. This structure resulted from the more tightly packed cells that did not undergo the delamination that characterized the other groups. In pig skin studies [16], it was found that exposure to AgNP at the level of 0.34 µg/mL was the cause of slight intracellular epidermal edema, while the level of 34 µg/mL influenced moderate intercellular and also intracellular epidermal edema. The level of 25 µg/mL used in our study did not induce any changes compared to the control group. The AgNP that we used were produced by a physical method, so we used high-purity silver electrodes immersed in ultrapure water. This method allowed us to obtain AgNP with a very high degree of purity, which avoids the effect of chemical contamination associated with the production of nanoparticles by chemical methods. Moreover, in our research, contrary to the research of Samberg et. al. in 2010 [16], we did not find the presence of nanoparticles in the surface layers of the epidermis, which suggests a high dependence of the process of skin penetration by AgNP on the exposure time, which in our experience was much shorter—only 1 h. Stratum corneum is the most important protective barrier of the skin, and the potential presence of nanoparticles in this zone may suggest further penetration of nanoparticles deep into the skin, which was observed after 40 h of exposure of patients’ skin to AgNP [63]. Nevertheless, the toxic effect of AgNP may be related not only to the time of exposition but also to the purity of nanoparticles.

Next is stratum granulosum [15]. Stratum corneum and stratum granulosum are most susceptible to potential damage. However, in the case of very irritating substances, the lower layers (spinous and basal layers) can also be degraded. In our study, not only in the stratum corneum but also in the stratum granulosum, no anomalies were found compared to the control group.

## 5. Conclusions

Summarizing our research, we can unequivocally confirm the suitability of the 3D skin model for nanotoxicity research. However, one may consider extending the duration of exposure to the tested nanostructures because their physical structure is a solid. The conducted studies showed that the biological effects of the ultrapure AgNP and GO are significantly different. Taking into account the analysis of protein expression of selected cytokines (IL-1α, IP-10, and MIP1β), it seems that GO may not have an indifferent effect. On the other hand, AgNP can be considered to be a pro-angiogenic factor. By analyzing the properties of the AgNP-GO complex in relation to the 3D epidermis model, no negative impact of the AgNP-GO complex was demonstrated on the survival of epidermal cells, as well as its morphological image, as assessed at the micro and ultrastructural level. Moreover, the assessment of genotoxicity and metabolism (MTT test) also did not indicate any potential harmfulness of the complex to the human epidermis. Moreover, key indicators of pro-inflammatory activity, such as the secretion of IL-1α into the medium, did not indicate a negative effect from the complex.

Analysis of the protein expression of some cytokines in epidermal tissue showed an increased level of selected pro-inflammatory cytokines such as IL-1α and RANTES. On the other hand, there was a decrease in highly pro-inflammatory MIP1β and an increase in anti-inflammatory TIMP-2 and TNF-α. In light of the obtained results, it seems that the AgNP-GO complex is not a completely neutral material, but it is also not a toxic material. The effect of the AgNP-GO complex can be described as activating the process of epidermis renewal, which makes it suitable for use as a material for skin contact.

## Figures and Tables

**Figure 1 pharmaceutics-14-01398-f001:**
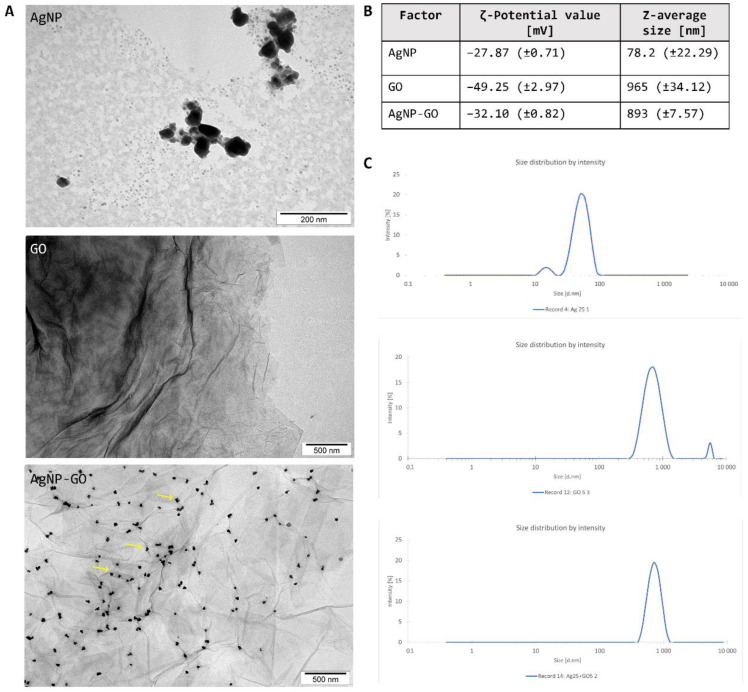
The characterization of the experimental factors. (**A**) The transmission electron microscopy (TEM) visualization of the AgNP, GO, and AgNP-GO complex. Examples of AgNP adhesion places are marked by yellow arrows. (**B**) Zeta potentials and Z-average size by dynamic light scattering analysis of the evaluated nanomaterials. (**C**) Size distributions of the analyzed hydrocolloids in water.

**Figure 2 pharmaceutics-14-01398-f002:**
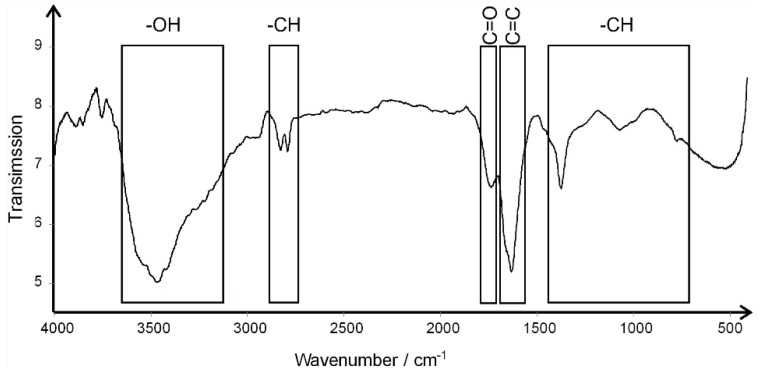
FTIR spectrum of silver nanoparticle and graphene oxide complex (AgNP-GO) registered in the middle region (4000–400 cm^−1^).

**Figure 3 pharmaceutics-14-01398-f003:**
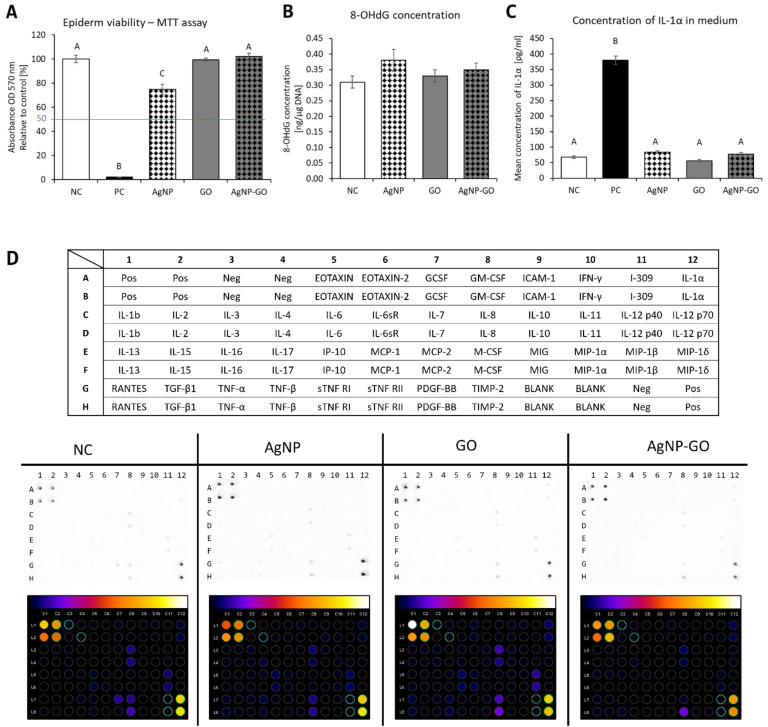
Biocompatibility of nanomaterials with Epiderm™. NC—negative control (DPBS), PC—positive control (5% SDS), AgNP—silver nanoparticles, GO—graphene oxide, AgNP-GO—complex. (**A**) EpiDerm™ viability measured by mitochondrial activity depletion. The MTT assay was performed 48 h after nanofactor treatments. The blue line represents the critical point at the level of 50%, below which the factor is recognized as an irritant. (**B**) The expression level of the oxidative DNA damage marker, 8-hydroxy-2’-deoxyguanosine (8-OHdG), in tissue as evaluated by enzyme-linked immunosorbent assays (ELISA). (**C**) The concentration level of the IL-1α released from Epiderm™ to the medium 24 h after experimental factor treatments, evaluated by a colorimetric ELISA test. The highest response was observed in the positive control (PC) group, which proves the functionality of the system. (**D**) Effect on the protein expression of proinflammatory cytokines in EpiDerm™ 24 h after treatment. The table represents a scheme of the protein pattern at the membrane. The results are normalized to the negative control (NC) group. Images were created with ImageJ software. The error bars represent standard error of mean (*n* = 4 per group). Different letters (A–C) above the columns indicate statistically significant differences (*p* ≤ 0.01) analyzed by Tukey’s HSD test.

**Figure 4 pharmaceutics-14-01398-f004:**
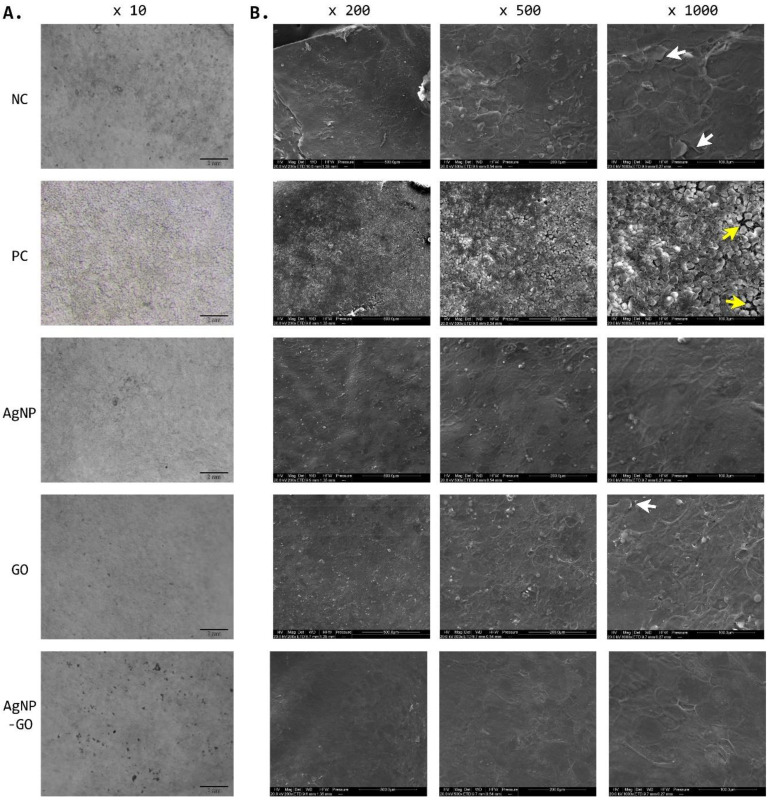
Visualization of EpiDerm™ surface 24 h after treatment. Comparison of silver nanoparticles (AgNP), graphene oxide (GO), and AgNP-GO complex activity to the negative control (NC, DPBS) and positive control (PC, SDS). (**A**) Digital photos of EpiDerm™ without fixation were obtained using a stereo microscope. (**B**) Tissues imaged in a scanning electron microscope were fixed in 2.5% l-glutaraldehyde in PBS. White arrows indicate the places of exfoliation and keratinization of the top layer; yellow arrows indicate a disruption of the continuity of the tissue surface.

**Figure 5 pharmaceutics-14-01398-f005:**
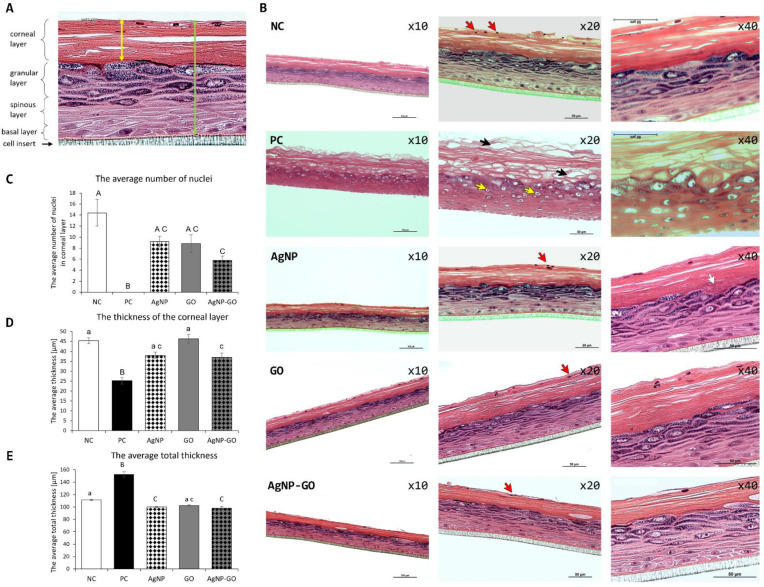
The EpiDerm™ morphology evaluated by optical microscopy after hematoxylin-eosin cross section staining. (**A**) Scheme of the EpiDerm™ structure. The double headed arrows indicate the boundaries of the measurement for the thickness of the corneal layer (yellow) and total thickness (green). (**B**) Visualization of the tissues after treatment at 10×, 20×, and 40× magnifications. Red arrows indicate the nuclei, black arrows indicate tissue damaged after SDS treatment, yellow arrows show the necrosis process, and the white arrow shows a place with local thickening of the stratum corneum. (**C**) The average number of nuclei in stratum corneum. (**D**) The average thickness of stratum corneum. (**E**) The average total thickness. Different letters (a, c) and (A–C) above the columns indicate statistically significant differences (*p* ≤ 0.05 and *p* ≤ 0.01 respectively) as analyzed by Tukey’s HSD test. NC—negative control, PC—positive control, AgNP—silver nanoparticles, GO—graphene oxide, AgNP-GO—complex of silver nanoparticles and graphene oxide.

**Figure 6 pharmaceutics-14-01398-f006:**
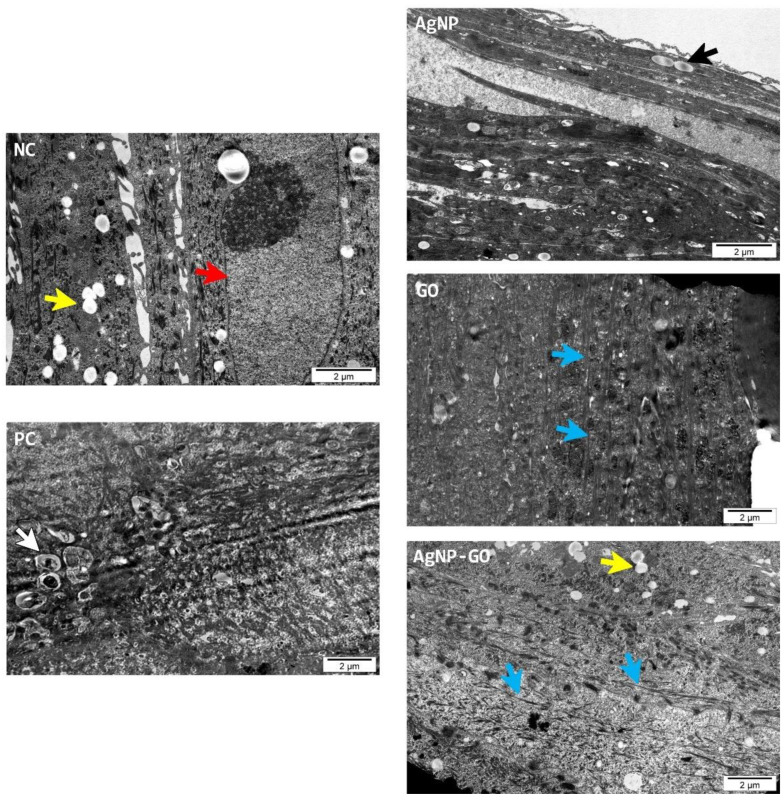
The EpiDerm™ ultrastructure evaluated with a transmission electron microscope (scalebar 2 µm). The arrows indicate the following: red—cells from the granular layer; yellow—lipid droplets; white—places of necrosis; black—nuclei in the stratum corneum; blue—bundle of keratin filaments.

## Data Availability

Data is contained within the article.

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
