# Peer review of "Molecular Biocompatibility of a Silver Nanoparticle Complex with Graphene Oxide to Human Skin in a 3D Epidermis In Vitro Model"

_pharmaceutics, 2022, doi:10.3390/pharmaceutics14071398_

Round 1

Reviewer 1 Report

In this manuscript, the authors reported about the molecular biocompatibility of AgNP with GR to human skin in a 3D epidermis in vitro model. Graphene Oxide-Ag nanoparticle is novel. Furthermore, the enthusiasm is dampened by lack of mechanistic insight. The characterization should have been more rigorous. There are few major issues that needs to be addressed before considering this manuscript for publication. 

1. In fig 1, why there is reduction in size when GO sheets are functionalized with AgNPs? What kind of interactions is happening between  AgNPs and GO complex?

2. In page 6 line 281, the authors claimed spherical morphology of AgNPs. However, the TEM images are showing irregular morphology of AgNPs.

3. Authors should performed studies regarding the pH stability of the material.

4. What is the release rate of AgNP from GO?

5. Cutaneous inflammation cannot be addressed in an in vitro set up in absence of immune cells. The translational relevance of the data provided in the manuscript is very weak. There is an active interaction between resident keratinocytes and visiting blood borne immune cells such as macrophages.

Reviewer 2 Report

Dear editor,

The manuscript (pharmaceutics-1761036) aims to report development of a hybrid material comprising Ag nanoparticles and GO and investigation of its biocompatibility via a 3D human skin model and MTT assay. Thanks to the cooperative effect of the constituents, the hybrid has been shown to provoke the renewal process of epidermis. The study is interesting and seems to be novel. However, before re-considering the work for publication, I have several major and minor comments that should be properly addressed:

1- Abstract; the first two sentences are non-scientific and meaningless.

2- Introduction, the first two paragraphs have no references! In general, the beginning part of introduction needs to be re-written. why the major shortcoming of disinfectants and decontaminating systems should be toxicity to human skin? Many basic sentences with no scientific message, e.g., "Nanomaterials, due to their physio-chemical properties, are promising antibacterial agents that are also highly biocompatible"!! Many statements need reference (e.g., page 2, line 85-87).

3- Page 6, were Ag NPs monodisperse (60 nm in diameter)? If not, standard deviation should be added to their size. DLS measurement shows a heterodisperse distribution of size, why it is so?

4- How Ag NPs bond with GO flakes? the type of physicochemical interaction should be discussed based on evidence such as ATR-FTIR.

5-  Why the size of complex is smaller than GO alone, according to the DLS measurement?

6- Figure 1C, the graphs are the original outputs of the machine. Please convert them to an origin/excell one.

Round 2

Reviewer 2 Report

Dear editor,

Given the applied modifications, the manuscript can be published.